# Catching SARS-CoV-2 by Sequence Hybridization: a Comparative Analysis

Alexandra Rehn,[a] Peter Braun,[a] Mandy Knüpfer,[a] Roman Wölfel,[a] Markus H. Antwerpen,[a] Mathias C. Walter[a]

[a]Bundeswehr Institute of Microbiology, Munich, Germany

**ABSTRACT** Controlling and monitoring the still ongoing severe acute respiratory syndrome coronavirus 2 (SARS-CoV-2) pandemic regarding geographical distribution, evolution, and emergence of new mutations of the SARS-CoV-2 virus is only possible due to continuous next-generation sequencing (NGS) and sharing sequence data worldwide. Efficient sequencing strategies enable the retrieval of increasing numbers of high-quality, full-length genomes and are, hence, indispensable. Two opposed enrichment methods, tiling multiplex PCR and sequence hybridization by bait capture, have been established for SARS-CoV-2 sequencing and are both frequently used, depending on the quality of the patient sample and the question at hand. Here, we focused on the evaluation of the sequence hybridization method by studying five commercially available sequence capture bait panels with regard to sensitivity and capture efficiency. We discovered the SARS-CoV-2-specific panel of Twist Bioscience to be the most efficient panel, followed by two respiratory panels from Twist Bioscience and Illumina, respectively. Our results provide on the one hand a decision basis for the sequencing community including a computation for using the full capacity of the flow cell and on the other hand potential improvements for the manufacturers.

**IMPORTANCE** Sequencing the genomes of the circulating SARS-CoV-2 strains is the only way to monitor the viral spread and evolution of the virus. Two different approaches, namely, tiling multiplex PCR and sequence hybridization by bait capture, are commonly used to fulfill this task. This study describes for the first time a combined approach of droplet digital PCR (ddPCR) and NGS to evaluate five commercially available sequence capture panels targeting SARS-CoV-2. In doing so, we were able to determine the most sensitive and efficient capture panel, distinguish the mode of action of the various bait panels, and compute the number of read pairs needed to recover a high-quality full-length genome. By calculating the minimum number of read pairs needed, we are providing optimized flow cell loading conditions for all sequencing laboratories worldwide that are striving for maximizing sequencing output and simultaneously minimizing time, costs, and sequencing resources.

**KEYWORDS** SARS-CoV-2, mutations, next-generation sequencing, NGS, enrichment, ddPCR, adaptive mutations, sequence capture

The world is still facing a tremendous and ongoing pandemic caused by a virus named severe acute respiratory syndrome coronavirus 2 (SARS-CoV-2). While the mere detection by reverse transcription-quantitative PCR (RT-qPCR) or antigen tests to confine the spread of this virus is a valuable diagnostic tool, next-generation sequencing (NGS) techniques were, are, and will be one of the keys to monitor and, hence, control this pandemic. Without the early availability of the SARS-CoV-2 genome (strain Wuhan-Hu-1) in January 2020 (1–3), the development of specific diagnostic RT-qPCR tests for the rapid detection of this virus would have been all but impossible (4). At

Address correspondence to Mathias C. Walter, mathiaswalter@instmikrobiobw.de.

present, next-generation sequencing plus sharing the sequence data via the GISAID initiative is the only way to monitor the geographical distribution of the circulating strains and the adaptation of the virus regarding its transmissibility (5–9), pathogenicity (10–12), and evolution (13, 14). Moreover, since antiviral treatments and vaccines have been developed against SARS-CoV-2, it is vital to know whether a newly emerged strain will develop resistance (15, 16) against antivirals or will acquire vaccine-escaping mutations (17–19).

However, direct NGS of human swab samples from COVID-19-positive patients can be very expensive, time-consuming, and challenging. Because swab samples contain predominantly human cells with only a minor proportion of virus particles, direct sequencing of patient material is prone to miss the low-abundance species, especially if no target enrichment strategies were applied prior to sequencing. At the moment, two different target enrichment approaches (20, 21) are mainly used around the world: tiling multiplex PCRs (22–26) and sequence hybridization by bait capture (27–29). While the amplicon-based approach is very fast, sensitive, and easy to handle, it can lead to sequencing gaps in the case of divergence between the target genome and the amplicon primers due to mutations of the virus and is, hence, inconsistent in the elucidation of new SARS-CoV-2 mutations. Targeted capture-based approaches, on the other hand, tolerate up to 10% to 20% of mismatches between the target sequence and the so-called bait, which is made of biotinylated, single-stranded RNA/DNA probes complementary to the target nucleic acids. Regarding the emergence of new SARS-CoV-2 mutations, we therefore see more certainty in using capture-based approaches. However, no evaluation of the various, commercially available capture bait panels has been conducted so far. We therefore set out to compare five different capture panels within three library preparation protocols in order to determine the most sensitive and most efficient one, thereby providing pivotal information for all sequencing laboratories in the world that are currently occupied with SARS-CoV-2 sequencing and the monitoring of new emerging mutations.

## RESULTS

**Experimental setup.** In order to determine the sensitivity and capture efficiency of the different bait panels (Illumina respiratory panel v1 and v2, MyBaits SARS-CoV-2 panel, and Twist Bioscience SARS-CoV-2 panel and respiratory panel), five RNA input pools, varying in the ratio of the concentrations of SARS-CoV-2 and human reference RNA (HRR) to simulate human RNA background in patient samples, were produced. Absolute concentrations of SARS-CoV-2 and human RNA were quantified by droplet digital PCR (ddPCR) using the targets ORF1a and human ubiquitin C (UBC), respectively. The ORF1a-to-UBC ratio was adjusted to $10^{-5}$ in pool 1, $10^{-4}$ in pool 2, $10^{-3}$ in pool 3, $10^{-2}$ in pool 4, and $10^{-1}$ in pool 5. The ratio of the produced input pools and the logarithmic change of the SARS-CoV-2 concentration were confirmed by ddPCR and reverse transcription-quantitative PCR (RT-qPCR) (see Fig. S1a and b in the supplemental material). Subsequently, all RNA input pools were subjected to reverse transcription and second-strand synthesis before they were entered into three different library preparation protocols provided by the companies Illumina, New England Biolabs (NEB), and Twist Bioscience (Fig. 1). Each library preparation was followed by an enrichment with a separate capture panel. In the case of the Illumina library preparation, respiratory panels v1 and v2 from Illumina were used for the enrichment. The NEBNext Library preparation was followed by sequence hybridization with the MyBaits SARS-CoV-2 panel, while the Twist Bioscience library preparation preceded the capture with the SARS-CoV-2-specific panel and the respiratory panel from Twist Bioscience (Fig. 1). The change in the ratio of SARS-CoV-2 to human background was quantified by ddPCR before and after the capture. Finally, all enriched pools were sequenced on an Illumina MiSeq instrument.

**Library preparation protocols differ significantly in quality and quantity of the processed library.** Examination of the quality and quantity of the libraries is crucial for the subsequent sequencing and, in this case, for the consecutive target enrichment. Figure 2 shows a comparison of the libraries generated by the three different protocols

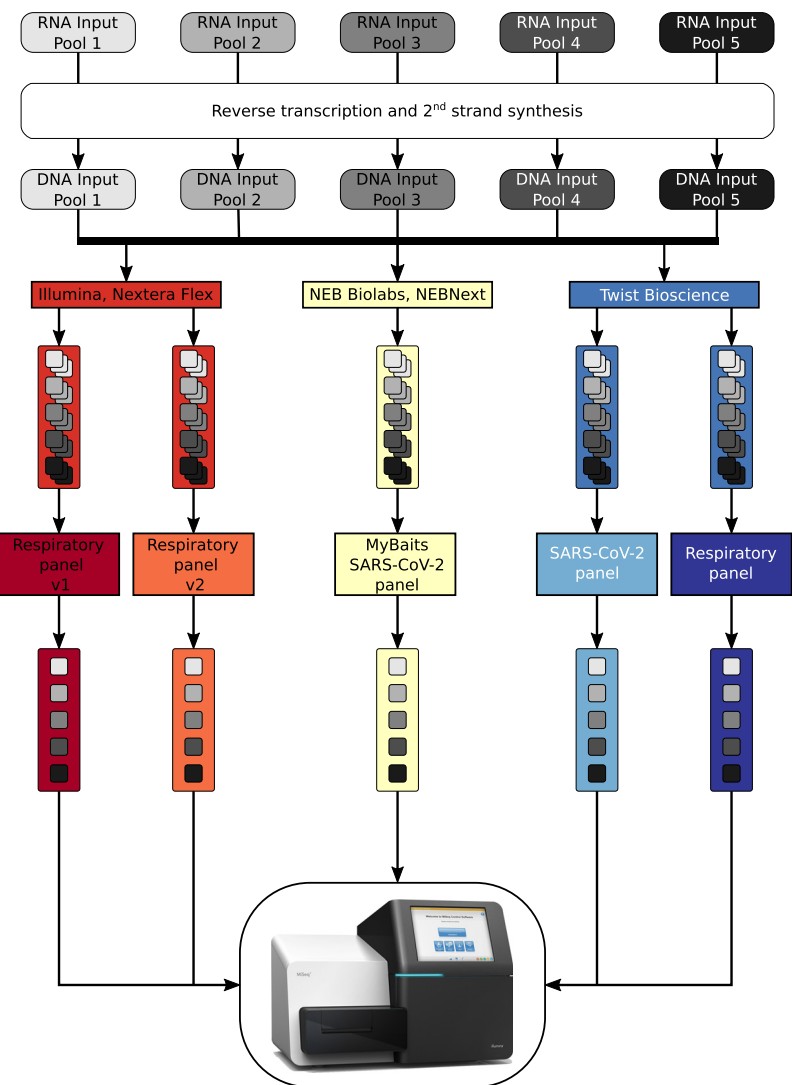

**FIG 1** Graphical overview of the performed workflow. RNA input pools, differing in SARS-CoV-2 concentration, were subject to reverse transcription and second-strand analysis before entering three different library preparation methods. For better statistics, each RNA input pool was used three times during each library preparation method and for each bait panel tested. Before sequence hybridization, the triplicates were pooled by mass, resulting in five pools per bait, which were sequenced on an Illumina MiSeq after the enrichment process.

with regard to fragment size, library concentration, and total library mass. In terms of the mean fragment size and distribution, the Illumina Nextera Flex protocol produced the longest fragments, with a mean length of about 600 bp, yet yielded the most atypical distribution, as a second peak was visible in all samples (Fig. S2). The libraries generated by the NEBNext and Twist Bioscience protocols resulted in mean fragment sizes of around 400 bp and 500 bp, respectively, and showed a typical Gaussian size distribution (Fig. 2a; Fig. S2). Of note, all methods produced comparable fragment sizes across pools 1 to 5, indicating highly reproducible procedures with a given input concentration. In contrast, the concentrations and thus the final library masses varied strongly between the three protocols (Fig. 2b and c). Here, the library preparation method of Twist Bioscience achieved the highest library concentrations, surpassing its competitors by factors of 1.4 and 7.5. Again, discrepancies between the pools were within the error range and indicate a stable and reproducible library preparation procedure for a given initial concentration. As single libraries ought to be pooled by mass prior to the sequence hybridization process according to the manufacturers' protocols, a comparison

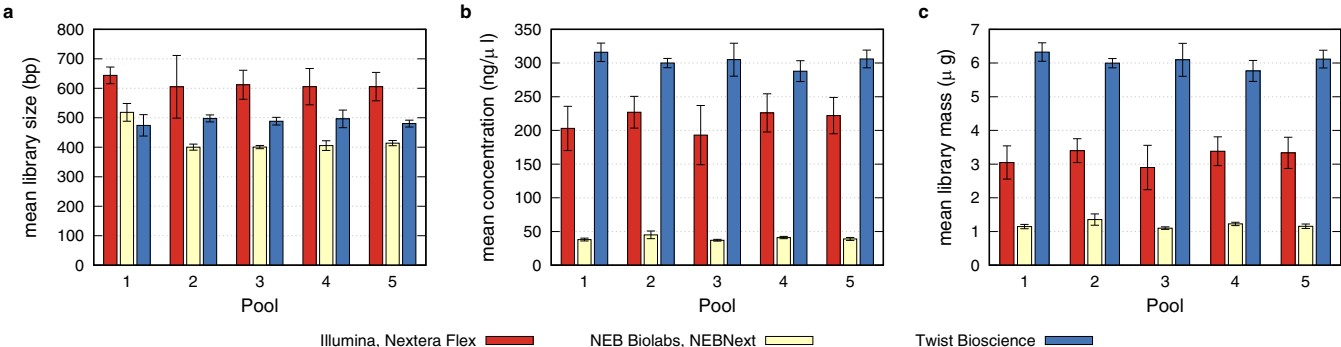

**FIG 2** Comparison of the quality control parameters after library preparation by three different methods. (a) Mean library size obtained by the analysis of the fragment size of the triplicates per input pool. Usage of the Illumina Nextera Flex protocol results in the largest libraries, followed by the libraries of Twist Bioscience and NEBNext. (b) Concentrations of the individual libraries were analyzed with a Qubit fluorometer. Combining the values of the triplicates per input pool resulted in a mean concentration per pool. Here, the libraries produced by the Twist Bioscience protocol reached the highest mean concentration, followed by the libraries of the Illumina Nextera Flex and the NEBNext protocols. (c) Mean library mass was determined by the measured concentration and the elution volume. Here again, the Twist Bioscience libraries succeeded those of the Illumina Nextera Flex and NEBNext.

of the final masses is beneficial (Fig. 2c). Due to the highest library concentrations and the second highest elution volume, the library preparation method provided by Twist Bioscience resulted in the highest final library masses available for the subsequent sequence hybridization capture.

**Capture bait panels differ in their affinities toward SARS-CoV-2.** In order to evaluate the sensitivity and capture efficiency of the five different bait panels, the triplicates originating from the same RNA input pools were pooled by mass and quantified by ddPCR before and after the sequence hybridization process (Fig. 3a and b). Primers targeting the open reading frame ORF1a were used to quantify the presence of SARS-CoV-2-specific library fragments, while UBC was used as a marker for human nontarget

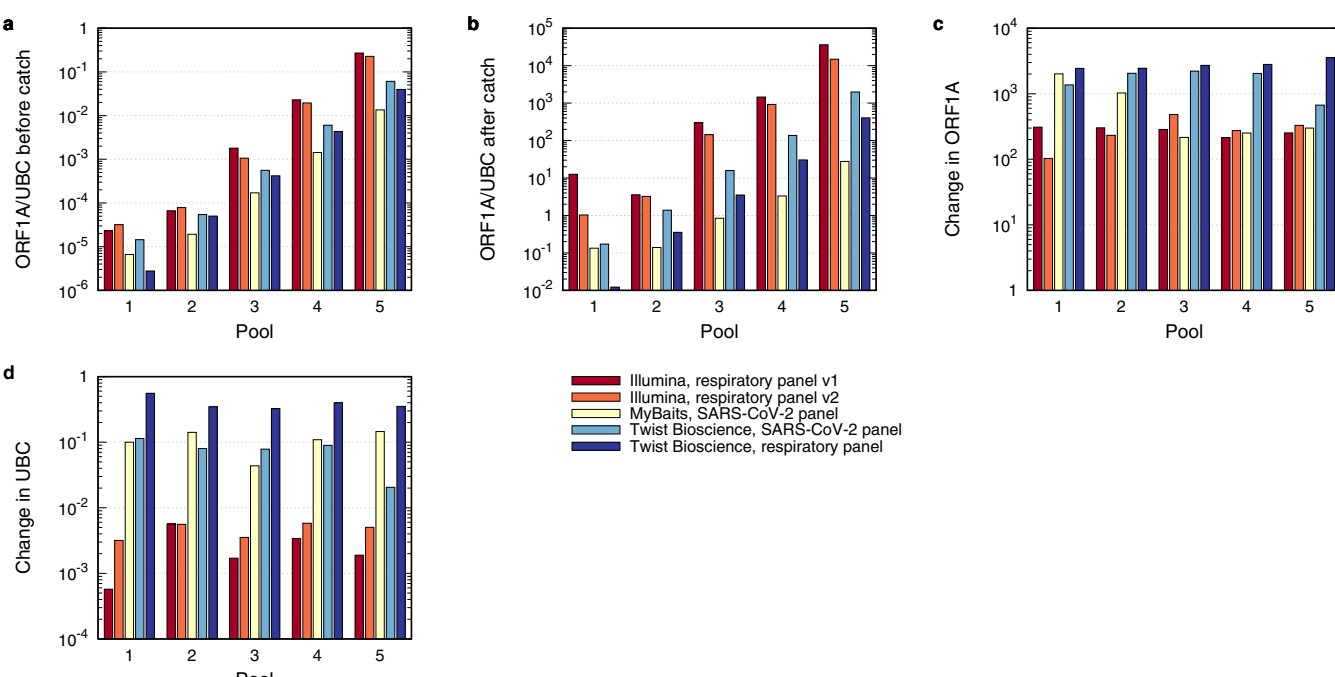

**FIG 3** Analysis of the hybridization sequence capture by ddPCR. (a and b) SARS-CoV-2-specific libraries were quantified by primers targeting ORF1a, while nontarget libraries were quantified by the presence of human ubiquitin C (UBC). The ORF1a/UBC ratio was plotted before (a) and after (b) the enrichment, with the highest ratio shown for the two Illumina panels, followed by the two Twist Bioscience panels and the MyBaits panel. (c and d) The change in ORF1a and UBC was plotted by dividing the counted concentration of ORF1a and UBC, respectively, after the enrichment with the respective concentration before the enrichment. The strongest change in ORF1a was observed by the two Twist Bioscience panels, while the strongest reduction in UBC was detected with the Illumina panels.

mSystems®

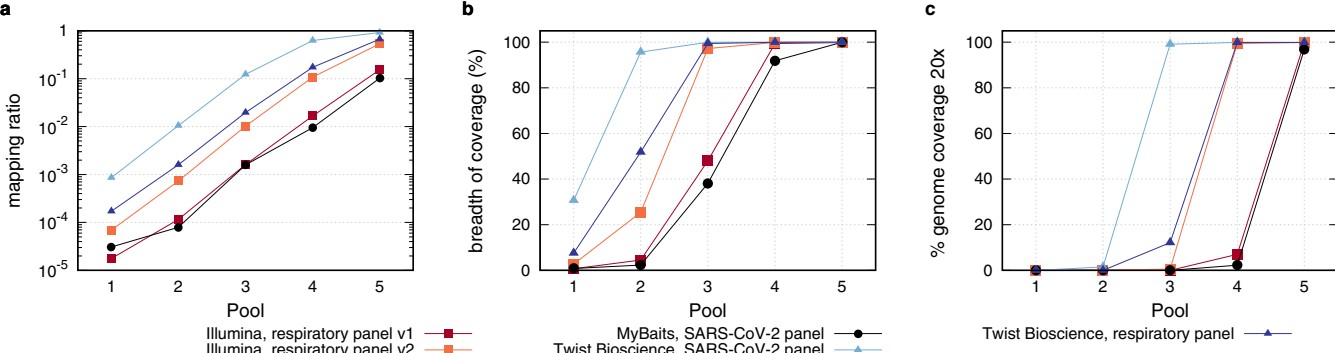

**FIG 4** Analysis of the efficiency of the sequence hybridization panels by NGS. (a) The numbers of SARS-CoV-2 mapping reads out of a subset of 130,000 reads were plotted against the pools, with the highest mapping ratio shown for the Twist Bioscience SARS-CoV-2 panel. (b) Breadth of coverage, defined by the number of covered bases of the SARS-CoV-2 genome, was compared for all panels. Use of the Twist Bioscience SARS-CoV-2 panel led to a nearly complete coverage of the SARS-CoV-2 genome already in pool 2, while the respiratory panels of Twist Bioscience and Illumina reached the full breadth of coverage in pool 3. (c) Comparison of the panels in regard to reaching a full-length genome with a coverage of 20×.

libraries. Preenrichment ORF1a/UBC ratios, depicted in Fig. 3a, reflect the exponential differences between the pools. Interestingly, the ORF1a/UBC ratio differed between the library preparation protocols, with the Illumina Nextera Flex protocol yielding the highest and the NEBNext libraries the lowest ORF1a/UBC ratio. The nature of this effect remains elusive so far and was not further addressed. Figure 3b shows the postenrichment ORF1a/UBC ratio. Again, the two Illumina panels showed the highest ORF1a/UBC ratio in all five pools, followed by the Twist Bioscience SARS-CoV-2 panel, the Twist Bioscience respiratory panel, and the MyBaits SARS-CoV-2 panel. Moreover, all bait panels still reflected the exponential gradation of the ORF1a/UBC ratios from one pool to the next. To discriminate the mode of action of the different bait panels during the enrichment process, the changes in ORF1a and UBC concentrations before and after the catch were compared using ddPCR (Fig. 3c and d). Here, the Illumina bait panels achieved an ORF1a and, hence, a SARS-CoV-2 enrichment of about 100-fold. This together with the strongest depletion of UBC (Fig. 3d) resulted in the highest ORF1a/UBC ratios after enrichment. Both panels from Twist Bioscience, on the other hand, yielded the strongest enrichment of ORF1a (Fig. 3c) but were not able to decrease the UBC concentrations by more than 1 order of magnitude, especially the respiratory panel (Fig. 3d). The MyBaits SARS-CoV-2 panel was efficient neither in the enrichment of ORF1a-specific sequences nor in the depletion of UBC.

**Comparison of sequence capture efficiencies.** After sequence hybridization, all enriched libraries were checked for concentration and fragment size (Fig. S3) and were subsequently sequenced on an Illumina MiSeq instrument. For an accurate comparison of the five different bait panels, all existing MiSeq reads were subsampled to 130,000 reads, which were previously corrected for PCR duplicates. All SARS-CoV-2 mapping reads within that subset were identified, and the SARS-CoV-2/nontarget ratio for each pool was plotted (Fig. 4a). Use of the SARS-CoV-2-specific bait panel of Twist Bioscience resulted in the highest abundance of SARS-CoV-2-specific reads in each pool, followed by the respiratory panel of Twist Bioscience and the Illumina respiratory panel v2, while the Illumina respiratory panel v1 and the MyBaits SARS-CoV-2 panel produced the lowest number of SARS-CoV-2-specific reads (Fig. 4a). Consistently, when the SARS-CoV-2-specific bait panel by Twist Bioscience was used, almost every base was already covered in pool 2 and resulted in a high-quality SARS-CoV-2 genome, in which every nucleotide of the genome was covered at least 20-fold, in pool 3 (Fig. 4b and c). This was one pool and, hence, 1 order of magnitude earlier than the respiratory panel of Twist Bioscience and the respiratory panel v2 of Illumina, which were themselves another order of magnitude better than the respiratory panel v1 of Illumina and the MyBaits SARS-CoV-2 panel (Fig. 4b and c). Summarizing the NGS data, the SARS-CoV-2-specific panel from Twist Bioscience was the most sensitive panel and showed the highest capture efficiency.

**TABLE 1** Number of reads needed to retrieve a full-length SARS-CoV-2 genome with a coverage of at least 20-fold with respect to enrichment panel used and the $C_T$ values of the input pools

| | | No. of reads | | | | |
|---|---|---|---|---|---|---|
| | | Illumina | | | Twist Bioscience | |
| Pool | $C_T$ | Respiratory panel v1 | Respiratory panel v2 | MyBaits SARS-CoV-2 | SARS-CoV-2 | Respiratory panel |
| 1 | 33.4 | 523,406,733[a] | 85,621,550[a] | 232,980,233[a] | 5,448,028 | 26,219,486 |
| 2 | 29.7 | 61,024,451[a] | 10,439,611 | 102,125,854[a] | 533,345 | 3,169,368 |
| 3 | 26.0 | 5,274,970 | 810,061 | 5,695,482 | 41,582 | 282,045 |
| 4 | 22.7 | 514,389 | 74,824 | 829,525 | 9,528 | 29,996 |
| 5 | 19.3 | 52,349 | 14,057 | 77,856 | 5,773 | 7,532 |

[a]The number of reads needed exceeds the maximum number of reads on an Illumina MiSeq v2 flow cell (30 million).

In order to analyze the minimum number of reads needed to retrieve a full-length SARS-CoV-2 genome with a coverage of at least 20-fold (Table 1), the ratio of all reads to those mapping to the SARS-CoV-2 genome was determined (Fig. S4), multiplied by the genome size, divided by the insert size, and corrected by the number of PCR duplicates. Table 1 shows that use of the SARS-CoV-2 panel from Twist Bioscience resulted in the lowest number of reads needed to recover a full-length SARS-CoV-2 genome. In fact, the number of reads was an order of magnitude lower than those of the respiratory panels from Twist Bioscience and Illumina, respectively, within the same pool. Again, the Illumina respiratory panel v1 and the MyBaits SARS-CoV-2 panel performed significantly less efficiently than those previously mentioned.

**Reasons for low capture rates and high nontarget ratio.** To evaluate the source of the high number of nontarget reads, especially in pools with a low input concentration of SARS-CoV-2, all nontarget reads of all pools within a specific capture panel were mapped. Table S1 shows a list of the top 30 hits of all panels, thereby revealing mainly rRNA targets when sorting by the number of total hits. These reads account for between 56% and 97% of all nontarget reads in the majority of panels, with the exception of the respiratory panel v1 of Illumina, in which rRNA causes only 7.3% of the hits (Table 2). Interestingly, the highest number of hits (about 25%) in this panel was assigned to GAPDH, which was drastically reduced in the successor version Illumina respiratory panel v2 and is obsolete in the capture panels of the other companies. We further analyzed the nature of nontarget reads in 240 sequenced patient samples (89 samples with Illumina respiratory panel v1, 108 samples with Illumina respiratory panel v2, 23 samples with Twist Bioscience, and 20 samples with MyBaits). Strikingly, the majority of nontarget reads were no longer rRNAs but revealed one long noncoding RNA (KCNQ1) and several mRNAs, especially titin (Tables S2 to S4). Moreover, all bait panels caught the same major nontarget hits, indicating specific bait sequences as the cause of the nontarget binding. Nevertheless, this analysis reveals a possible option for a further improvement in the capture bait panels.

**TABLE 2** Number of nontarget reads and their major hits

| | Value for: | | | | |
|---|---|---|---|---|---|
| | Illumina | | | Twist Bioscience | |
| Panel | Respiratory panel v1 | Respiratory panel v2 | MyBaits SARS-CoV-2 | SARS-CoV-2 specific | Respiratory panel |
| No. of nontarget reads | 2,527,450 | 1,896,363 | 4,817,333 | 1,286,780 | 2,186,252 |
| No. of rRNA reads | 183,994 | 1,066,572 | 3,010,428 | 811,090 | 2,132,833 |
| % nontarget rRNA reads | 7.3 | 56.2 | 94.1 | 63.0 | 97.6 |
| No. of GAPDH reads | 630,000 | 176,046 | 0 | 0 | 0 |
| % nontarget GAPDH reads | 24.9 | 9.3 | | | |

**TABLE 3** Overview of bait panel characteristics

| Panel | Bait length (nt) | Tiling | No. of targeted viruses | No. of PCR cycles preenrichment | No. of PCR cycles postenrichment |
|---|---|---|---|---|---|
| Illumina respiratory panel v1 | 80 | NA$^a$ | 41 | 12 | 12 |
| Illumina respiratory panel v2 | 80 | NA | 41 | 12 | 12 |
| MyBaits SARS-CoV-2 panel | 80 | 3× | 1 | 13 | 14 |
| Twist Bioscience SARS-CoV-2 panel | 120 | 4× | 1 | 12 | 16 |
| Twist Bioscience respiratory panel | 120 | 4× | 29 | 12 | 16 |

$^a$NA, information not available.

## DISCUSSION

The SARS-CoV-2 pandemic, which originated in Wuhan in December 2019, is still ongoing and reached new records of infected individuals in December 2020 despite the use of numerous counteractive measures. Since the beginning of the pandemic, whole-genome sequence data generated by next-generation sequencing were shared publicly on platforms like GISAID and played a pivotal role in the identification (1), the development of diagnostic (4) and therapeutic (30, 31) strategies, and the investigation of the origin (13) and the evolution of the virus. Driven by the appearance of potentially more aggressive, more infectious, or immunity-escaping strains like B.1.1.7 (United Kingdom) (6, 32), B.1.351 (South Africa) (7, 18), and P1 (alias of B.1.1.28.1, Brazil) (33, 34), the World Health Organization (WHO) initiated a sequencing program (35) in January 2021 to monitor the virus's movement, activity, and evolution with its impact on transmissibility, pathogenicity, and immunity. In order to reach these goals, a large number of SARS-CoV-2 genomes will need to be sequenced continuously and efficiently in terms of time and costs. Therefore, target enrichment protocols like capture-based or amplicon-based approaches are inevitable and allow for more samples to be sequenced in parallel (36). While the amplicon-based approach (ARTIC [23] and follow-up designs by various suppliers), which generates target amplicons from 400 to 2,000 bp, is very sensitive, it is also more prone to amplicon failure due to divergences in the target genome at the primer binding sites, leading to gaps in the genome sequence and, hence, loss of potentially important information, especially when looking for new mutations. Targeted capture-based approaches, on the other hand, are able to tolerate up to 10% to 20% of mismatches between the target sequence and the bait panel (35), thereby providing a stable technique in the monitoring of new SARS-CoV-2 variants. We therefore set out to compare five different capture bait panels with regard to their sensitivity and capture efficiency toward SARS-CoV-2 by a combined approach of ddPCR and NGS. Of note, we performed these experiments with only one variant of SARS-CoV-2, knowing that the occurrence of point mutations and even small deletions will not lead to a performance loss, as the used bait panels are at least 80 nucleotides long and have a tiling of at least 3× (Table 3).

Our results demonstrate that all tested bait panels were able to bind SARS-CoV-2 libraries but showed great differences in sensitivity and enrichment capacity. Overall, the SARS-CoV-2 panel of Twist Bioscience performed as the most sensitive and the most efficient capture panel, followed by the respiratory panel from Twist Bioscience, the respiratory panel v2 from Illumina, its progenitor respiratory panel v1, and the MyBaits SARS-CoV-2 panel. We speculate that this hierarchy is a result of the combination of three parameters: first, the enrichment factor for ORF1a/SARS-CoV-2 reads; second, the depletion factor for nontarget reads; and finally, yet importantly, the fragment size after the sequence hybridization. The SARS-CoV-2-specific panel from Twist Bioscience showed together with the respiratory panel from Twist Bioscience the highest enrichment factor for ORF1a/SARS-CoV-2 but exceeded the respiratory panel in the depletion of UBC/nontarget reads (Figure 3c and d). Additionally, all Twist Bioscience libraries displayed the largest postenrichment fragment size (Fig. S3b and Table S6), thereby rendering both panels as the best and second-best performing ones. Notably, the PCR duplication ratio was the highest for the Twist Bioscience SARS-CoV-2-specific panel and the lowest for the Twist Bioscience respiratory panel (Table S6). The Illumina

respiratory panels v1 and v2, on the other hand, showed only an enrichment factor for ORF1a/SARS-CoV-2 of about 100-fold but performed best in the depletion of the UBC/nontarget reads (Fig. 3c and d). Nevertheless, the postenrichment fragment size of the Illumina libraries was significantly smaller than that from Twist Bioscience (Fig. S3b and Table S6), and the PCR duplication ratio was at 13% on average. NGS data of the Illumina respiratory panel v2 showed a higher number of target reads (Fig. 4a), thereby surpassing the older version, respiratory panel v1. The MyBaits SARS-CoV-2 panel was the only capture-based approach sold as a stand-alone product without any recommended library preparation protocol. Here, we observed that the combination of the NEBNext Ultra II library preparation protocol and the MyBaits SARS-CoV-2 panel resulted in the least sensitive combination with the lowest capture efficiency. Our data clearly revealed that the NEBNext library preparation resulted in the shortest libraries with the lowest concentrations (but with a duplication ratio of only 6.2% on average). Whether this was the main cause for the poor performance or if it was the combination of this bait panel with this library preparation is impossible to tell from our data, since the combination of the best-performing Twist Bioscience SARS-CoV-2 bait panel with the NEBNext library was not performed.

Looking at the PCR duplication ratio in depth, we cannot find any correlation between PCR duplication ratio (Table S6) and number of PCR pre- and postenrichment cycles (Table 3). The PCR duplication ratio correlates only with the number of target reads and decreases with lower threshold cycle ($C_T$) values. Since all the Illumina and Twist Bioscience panels were sequenced with the same run, it is unlikely that optical duplicates are the main cause. We can only speculate that either biological duplicates or reverse transcriptase/fragmentation/ligation bias during library construction are the reason for the different duplication ratios.

To date, this is the first study comparing capture enrichment panels for SARS-CoV-2. We were able to identify the best-performing one and successfully deconstructed the mode of SARS-CoV-2 enrichment and depletion of nontarget reads between the different panels. By combining this information, we are proposing on the one hand an improvement in the capture efficiency by removing individual bait sequences that are responsible for catching off-target molecules. On the other hand, our study provides a correlation between the SARS-CoV-2 concentration (measured by RT-qPCR or ddPCR) and the minimal number of reads needed to recover a high-quality full-length genome. This information can be applied in all sequencing labs worldwide for calculation of the maximum number of patient samples being loaded onto a single flow cell, thereby reducing valuable time, sequencing resources, and costs. Hence, this work may pave the way for high-throughput yet high-quality screening for the worldwide emerging new mutations of SARS-CoV-2 and hence contribute to a more effective containment of the ongoing COVID-19 pandemic.

## MATERIALS AND METHODS

**Cultivation and purification of SARS-CoV-2.** SARS-CoV-2 virus (derived from a patient sample, lineage B1) was cultured in Vero E6 cells with minimal essential medium (MEM) containing 2% fetal bovine serum (FBS) at 37°C with 5% $CO_2$ and was harvested 72 h postinfection. Virus stocks were stored at −80°C. Viral RNA was extracted using diatomaceous earth (37). Briefly, 140 $\mu$l of virus-containing supernatant was added to 560 $\mu$l lysis buffer (800 mM guanidine hydrochloride, 50 mM Tris [pH 8.0], 0.5% Triton X-100, 1% Tween 20) and incubated at room temperature for 10 min. Subsequently, 560 $\mu$l ethanol (VWR, Darmstadt, Germany) and 20 $\mu$l diatomaceous earth (100 mg/ml in distilled water; VWR, Darmstadt, Germany) were added to the mixture. After vigorous vortexing, the diatomaceous earth-cell culture mixture was incubated at room temperature for 5 min with shaking to prevent sedimentation of the diatomaceous earth. After centrifugation at 13,000 rpm for 3 min at room temperature, the supernatant was discarded. A 500-$\mu$l volume of washing buffer (10 mM Tris [pH 8.0], 0.1% Tween 20) was added, and the mixture was centrifuged at 13,000 rpm for 3 min. After the supernatant was discarded, 500 $\mu$l of washing buffer (10 mM Tris [pH 8.0], 0.1% Tween 20) was added, and the mixture was centrifuged again for 3 min at 13,000 rpm. After the supernatant was decanted, 400 $\mu$l of acetone (Roth, Karlsruhe, Germany) was added to the pellet, vortexed, and centrifuged again. After removal of the supernatant, the pellet was dried for 5 min at 56°C and the viral RNA was eluted with 80 $\mu$l of distilled water. After mixing and centrifugation, the RNA was transferred to a new reaction tube and stored at −80°C until further use.

**TABLE 4** Primers and probes used in this study

| Name | Sequence 5′→3′ | Reference |
|---|---|---|
| UBC forward | ATTTGGGTCGCGGTTCTTG | 46 |
| UBC reverse | TGCCTTGACATTCTCGATGGT | 46 |
| UBC probe | FAM-TCTGACTGGTAAGACCATCACCCTCG-BHQ1 | 46 |
| Noblis.12 forward | ACGGCAGTGAGGACAATCAG | 47 |
| Noblis.12 reverse | CTGCAACACCTCCTCCATGT | 47 |
| Noblis.12 probe | HEX-CCAACAGTGGTTGTTAATGCAGCCA-BHQ1 | 47 |

**Quantification of SARS-CoV-2 and human RNA and cDNA by ddPCR and RT-ddPCR.** For quantification of human ubiquitin C mRNA (UBC) and SARS-CoV-2 ORF1a RNA, 20 μl droplet digital PCR (ddPCR) mix consisted of 5 μl One-Step RT-ddPCR advanced supermix for probes (Bio-Rad Laboratories, Munich, Germany), 2 μl of reverse transcriptase (Bio-Rad Laboratories, Munich, Germany; final concentration, 20 U/μl), 1 μl of dithiothreitol (DTT) (Bio-Rad Laboratories, Munich, Germany; final concentration, 15 nM), 1 μl 20× UBC primer and probe mix (Table 4) (final concentrations: primers, 900 nM; probe, 250 nM), 1 μl of 20× ORF1a primer and probe mix (Table 4) (final concentrations: primers, 900 nM; probe, 250 nM), 5 μl of nuclease-free water (Qiagen, Hilden, Germany), and 5 μl of template RNA. Partitioning of the reaction mixture into up to 20,000 droplets was carried out on a QX200 ddPCR droplet generator (Bio-Rad Laboratories, Munich, Germany), and PCR was performed using a Mastercycler Pro (Eppendorf, Wesseling-Berzdorf, Germany) with the following thermal protocol. Reverse transcription (RT) was performed at 50°C for 60 min. An enzyme activation step at 95°C was carried out for 10 min, followed by 40 cycles of a two-step program of denaturation at 94°C for 30 s and annealing/extension at 58°C for 1 min. Final enzyme inactivation was performed at 98°C for 10 min. Finally, the samples were cooled down to 4°C. All steps were performed using a temperature ramp rate of 2°C/s. Afterwards, PCR droplets were analyzed using a QX100 droplet reader (Bio-Rad Laboratories, Munich, Germany), and QuantaSoft Pro software (Bio-Rad Laboratories, Munich, Germany) was used for absolute quantification of target concentrations.

When cDNA was used as a template, the 20-μl ddPCR mix consisted of 10 μl ddPCR supermix for probes (Bio-Rad Laboratories, Munich, Germany), 1 μl 20× UBC primer and probe mix (Table 4) (final concentrations: primers, 900 nM; probe, 250 nM), 1 μl of 20× ORF1a primer and probe mix (Table 4) (final concentrations: primers, 900 nM; probe, 250 nM), 3 μl of nuclease-free water (Qiagen, Hilden, Germany), and 5 μl of template containing cDNA. Subsequent steps were carried out as described for RT-ddPCR, with the difference that no initial reverse transcription step was included in the thermal cycling protocol.

**Generation of RNA input pools.** In order to create RNA pools with various SARS-CoV-2 concentrations, the initial concentrations of purified SARS-CoV-2 and the universal human reference RNA (UHRR; Agilent Technologies, product number 740000) were determined by ddPCR as described above. Subsequently, each RNA input pool was calculated to have a SARS-CoV-2-to-UBC ratio of $10^{-5}$ in pool 1, $10^{-4}$ in pool 2, $10^{-3}$ in pool 3, $10^{-2}$ in pool 4, and $10^{-1}$ in pool 5. Evaluation of the SARS-CoV-2/UBC ratio of these RNA input pools was again done by ddPCR.

**Reverse transcription and second-strand synthesis.** Depending on the subsequent library preparation protocol, two different reverse transcriptases were used. In the case of Illumina Nextera Flex and NEB NEBNext, SuperScript IV (Thermo Fisher Scientific, Langerwehe, Germany) was applied according to the manufacturers' recommendations, while ProtoScript II (New England Biolabs, Frankfurt am Main, Germany) was used for the Twist Bioscience workflow according to the details given in the Twist Bioscience library preparation protocol. To improve the efficiency of all reverse transcriptases, the random hexamers (random primer 6 in the case of ProtoScript II) were mixed with 10% oligo(dT) primers. In all cases, the NEBNext Ultra II nondirectional RNA second-strand synthesis buffer and reagents (New England Biolabs, Frankfurt am Main, Germany) was used for the second-strand synthesis.

**Library preparation.** Library preparation was performed according to the manufacturers' protocols. For Illumina, the Nextera Flex for Enrichment (v03) was used with the following deviations: in the step "amplify tagmented DNA," the initial denaturation time was prolonged from 3 min to 4 min. Furthermore, the denaturation time during the 12 cycles of amplification was set to 30 s instead of 20 s. For the preparation of the Twist Bioscience libraries, the guide "Creating cDNA Libraries using Twist Library Preparation Kit for ssRNA Virus Detection" (version: August 2020) was followed according to the instructions given. In step 3.1, the fragmentation time was reduced from 22 min to 1 min. For NEBNext libraries, the manual "NEBNext Ultra II FS DNA Library Prep Kit for Illumina" was used. Here, we followed the instructions of section 1 for inputs of ≤100 ng but reduced the fragmentation time to 1 min.

**Sequence capture by hybridization.** In order to compare the five bait panels (Illumina respiratory panels v1 and v2, MyBaits SARS-CoV-2 panel, Twist Bioscience SARS-CoV-2 panel and respiratory panel), 200 ng of each triplicate was pooled and subsequently hybridized according to the manufacturer's instruction. In the case of the respiratory bait panels v1/v2 from Illumina, the hybridization was performed at 58°C and overnight. After washing, the enriched libraries were amplified for 12 cycles. Here, the initial denaturation time was prolonged to 60 s, while the denaturation during the cycles was set to 20 s. For enrichment of the Twist Bioscience libraries with either the SARS-CoV-2 specific or the respiratory panel, the manual "Twist Target Enrichment Protocol" was followed without any exception. Similarly, the MyBaits "Hybridization Capture for Targeted NGS" manual (version 4.01) was used according to the manufacturer's instructions to enrich the NEBNext libraries.

**Quality control of libraries and sequencing.** After library preparation and after the enrichment, the libraries had to pass a quality control check regarding concentration and size. The concentrations of the libraries were measured on a Qubit 4 fluorometer using the double-stranded DNA (dsDNA) HS assay kit (Thermo Fisher Scientific, Langerwehe, Germany). The shape and the mean fragment size of the libraries were determined on a model 5200 fragment analyzer using the HS NGS fragment kit (1 to 6,000 bp) (both from Agilent Technologies GmbH, Ratingen, Germany). Enriched libraries were loaded with a final concentration of 10 pM on a MiSeq flow cell using v3 reagent chemistry for 2 × 150 cycles (Illumina, Berlin, Germany).

**Data analysis.** Sequenced reads were cleaned from PCR duplicates using clumpify from the BBTools package (38) prior to subsampling them to 130,000 reads using seqtk (39) to get normalized data sets for each pool. Afterwards, subsampled reads were mapped against the SARS-CoV-2 Wuhan-Hu-1 reference genome sequence (1) with GenBank accession no. MN908947.3 using bwa mem (40). The number of mapped reads was determined using samtools flagstat (41), and coverage information was obtained using bedtools genomecov (42). The number of PCR duplicates was extracted from the clumpify log files. Data collection and overall statistics were generated using custom bash and awk scripts. Datamash (43) was used to aggregate the triplicate data sets, and gnuplot (44) was used for plotting.

To get a near-optimal pool ratio in correlation with the library concentration, we estimated the number of fragments needed for covering a full-length SARS-CoV-2 genome at a minimum of 20-fold by simply dividing the genome length (30,000 nucleotides) by the median mapping ratio of the pool and multiplying that number by the target coverage of 20×. The result was further corrected by the number of observed PCR duplicates (see Table S6 in the supplemental material) and multiplied by 2 to get the number of paired reads.

To investigate the high number of reads not mapping to the SARS-CoV-2 genome, a combined FASTA file containing all human reference genome sequences and the SARS-CoV-2 reference genome sequence as well as all annotated RNAs (noncoding and mRNAs) of both genomes was created. Then, Salmon (45) was used with default settings (k-mer = 31) to quantify the transcript abundance of all sequenced reads of each triplicate against this data set. Transcripts targeted by more than 100 reads were extracted and aggregated for each pool. For the 30 top most-targeted transcripts, their gene name and function were looked up and the transcripts were further aggregated if they belonged to the same gene.

**Data availability.** The data are available under BioProject accession number PRJNA717396, and the SARS-CoV-2 genome sequence is available at GISAID under accession number EPI_ISL_2699221.

## SUPPLEMENTAL MATERIAL

Supplemental material is available online only.

**FIG S1**, PDF file, 0.02 MB.
**FIG S2**, PNG file, 0.19 MB.
**FIG S3**, PDF file, 0.02 MB.
**FIG S4**, PDF file, 0.02 MB.
**TABLE S1**, DOCX file, 0.02 MB.
**TABLE S2**, DOCX file, 0.02 MB.
**TABLE S3**, DOCX file, 0.02 MB.
**TABLE S4**, DOCX file, 0.02 MB.
**TABLE S5**, DOCX file, 0.02 MB.
**TABLE S6**, DOCX file, 0.02 MB.

## ACKNOWLEDGMENTS

The project was partially funded by the BMBF-ZooSeq (grant no. 01KI1905A).

We thank Rahime Terzioglu, Peter Molkenthin, and Josua Zinner for excellent assistance with the ddPCR, RT-qPCR, and library preparation.

A.R., M.H.A., and M.C.W. developed the study design. M.H.A. and R.W. acquired funding and provided resources. A.R., P.B., and M.K. performed the experiments, and A.R., P.B., and M.C.W. analyzed the data. A.R. and M.C.W. wrote the manuscript, and P.B., M.K., R.W., and M.H.A. reviewed and edited it.

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
