## [Reviewer comments · mSystems]

Catching SARS-CoV-2 by sequence hybridization: a comparative analysis

Alexandra Rehn, Peter Braun, Mandy Knüpfer, Roman Wölfel, Markus Antwerpen, and Mathias Walter

Corresponding Author(s): Mathias Walter, Bundeswehr Institute of Microbiology

Review Timeline:

Submission Date:	April 1, 2021
Editorial Decision:	May 25, 2021
Revision Received:	July 8, 2021
Accepted:	July 12, 2021

Editor: Rachel Mackelprang

Reviewer(s): Disclosure of reviewer identity is with reference to reviewer comments included in decision letter(s). The following individuals involved in review of your submission have agreed to reveal their identity: Laurence Josset (Reviewer #2)

Transaction Report:

DOI: <https://doi.org/10.1128/mSystems.00392-21>

May 25, 2021

Dr. Mathias C. Walter
Bundeswehr Institute of Microbiology
Munich
Germany

Re: mSystems00392-21 (Catching SARS-CoV-2 by sequence hybridization: a comparative analysis)

Dear Dr. Mathias C. Walter:

Thank you for submitting your manuscript to mSystems. We have completed our review and I am pleased to inform you that, in principle, we expect to accept it for publication in mSystems. However, acceptance will not be final until you have adequately addressed the reviewer comments. I note that Reviewer #2 indicated that main flaw of the study is the use of RNA pools with only 1 isolate of SARS-CoV-2 instead of patient's samples with large diversity of SARS-CoV-2. While this appears to be a valid point, due to the time-sensitivity of this research, I think it is of benefit to forgo additional experiments in favor of publishing this study more rapidly. However, I suggest that the authors address this in the discussion.

Thank you for the privilege of reviewing your work. Below you will find instructions from the mSystemseitorial office and comments generated during the review.

Preparing Revision Guidelines

For complete guidelines on revision requirements, please see the Instructions to Authors at <https://msystems.asm.org/sites/default/files/additional-assets/mSys-ITA.pdf>. **Submissions of a paper that does not conform to mSystems guidelines will delay acceptance of your manuscript.**

Corresponding authors may join or renew ASM membership to obtain discounts on publication fees.

Need to upgrade your membership level? Please contact Customer Service at Service@asmusa.org.

Sincerely,

Rachel Mackelprang

Editor, mSystems

Journals Department
Reviewer comments:

Reviewer #1 (Comments for the Author):

The manuscript of Rehn and colleagues deals with comparing five different capture bait panels specific for SARS-CoV-2 towards their sensitivity and specificity by a combined approach of ddPCR and high-throughput sequencing. The general topic is timely and might be important for scientists who are interested in acquiring SARS CoV-2 full genomes. The study is well designed and the manuscript in general is sound and well written. A revised version of the paper might be advantageous for the reader.

In general, I suggest distinguishing between enrichment approaches (i.e., target-capture) and amplification methods (i.e., PCR) in the text. PCR methods are not typically designated as enrichment but as amplification methods. Regarding authors' aim to evaluate/determine sensitivity and specificity of the bait panels, I miss at least a clear conclusion about sensitivity and specificity of the bait panels.

In the abstract, the authors wrote that they "focused on optimization". In my opinion, the focus of the study is only on evaluation and not on optimization even though authors recommended some points for improvement. BTW, the published study should be forwarded to the companies developing the bait panels.

In the introduction, authors wrote "probes complementary to the target DNA". I suggest to add cDNA as well (or replace DNA by nucleic acids).

Also in the intro, the mention of the bait panel names is redundant.

Please, write "bait panels" or "capture panels" instead of just "baits" (throughout the ms) and ORF1a instead of 1A.

In the discussion, please add a dot in naming the SARS-Cov-2 lineages (e.g., B.1.1.7), and correct B.1.315 to B.1.351. What does "and Co." mean there?

Please arrange the table columns in Table 2 accordingly to Table 1 (as results shown in the Figures).

The manuscript should be reworked regarding comma/blank placement (e.g., first sentence of the second introductions' paragraph), especially Table 1 (i.e., use commas as separators for the read numbers).

In the methods, please add cities and countries for VWR, Roth, Thermo Fisher etc. as done for the other suppliers. In the "Sequence Capture by Hybridization" paragraph, please add the correct names of all tested bait panels (it makes more sense to mention them here than in the intro). Regarding supplementary Figures, the legends are very sparse and contain typos, legend for Fig. S4 is missing.

Reviewer #2 (Comments for the Author):

This technical paper compares the performances of five different capture baits for SARS-CoV-2 sequencing. Such comparison has not been done yet and is important for laboratories seeking the best method to sequence SARS-CoV-2.

However, the main flaw of the study is the use of RNA pools with only 1 isolate of SARS-CoV-2 instead of patient's samples with large diversity of SARS-CoV-2. The analysis of non-targets reads show that these pools do not necessarily reflect real patient samples, with non-targets reads being mostly rRNA in pools but not in patients samples. RNA extracted from patient's samples have usually low RIN (~2). What was the RNA quality of input pool? Does RNA quality impact the results? More importantly, there are no details about the SARS-CoV-2 isolate used for the pools. Where does it come from? Which lineage does it belong to? In the introduction and the discussion, the authors argue that capture-based approaches perform better than amplicon-based approaches for divergent viruses, however the paper does not evaluate whether this is similarly true for all tested baits. i.e do all baits capture equally well divergent SARS-CoV-2?

Other comments:

Fig 3 and 4 : Please add error bars and statistical analysis to show whether capture efficiency are reproducible within the triplicates?

Fig 4 and page 5: The analysis are made on reads "corrected for PCR duplicates". What is the impact of removing PCR duplicates? This is not done systematically in targeted sequencing.

Table 1: Number of reads needed for covering full-length SARS-CoV-2 at 20X should be calculated using minimal coverage depth, rather than median coverage depth.

Table 1: it seems that 1 asterisk is missing for Pool 2 result using Twist Respiratory Panel

Reviewer comments:

Reviewer #1 (Comments for the Author):

The manuscript of Rehn and colleagues deals with comparing five different capture bait panels specific for SARS-CoV-2 towards their sensitivity and specificity by a combined approach of ddPCR and high-throughput sequencing. The general topic is timely and might be important for scientists who are interested in acquiring SARS CoV-2 full genomes. The study is well designed and the manuscript in general is sound and well written. A revised version of the paper might be advantageous for the reader.

In general, I suggest distinguishing between enrichment approaches (i.e., target-capture) and amplification methods (i.e., PCR) in the text. PCR methods are not typically designated as enrichment but as amplification methods.

We agree with the reviewer, even though the WHO lists "Targeted amplicon-based approaches" under "Enriching SARS-CoV-2 genetic material prior to library preparation" (see "Genomic sequencing of SARS-CoV-2: A guide to implementation for maximum impact on public health", 08.01.2021). We did our best to change it throughout the manuscript wherever applicable.

Regarding authors' aim to evaluate/determine sensitivity and specificity of the bait panels, I miss at least a clear conclusion about sensitivity and specificity of the bait panels.

First of all, we decided to replace specificity with capture efficiency, as this term is more appropriate in the context of capture panels. We emphasized the results concerning the sensitivity and capture efficiency mainly in the NGS part within the results section but also in the discussion.

In the abstract, the authors wrote that they "focused on optimization". In my opinion, the focus of the study is only on evaluation and not on optimization even though authors recommended some points for improvement.

In the revised version, we try to clarify this point. When it comes to the library preparation workflow and the bait panels, we are focusing only on the evaluation of the methods. However, with the bioinformatics analysis, namely, the calculation of the number of read pairs needed to retrieve a full-length high quality genome and the determination of the off-target sequences; we provide optimization strategies for the users and the suppliers. The users benefit from the application of the calculation of the minimum number of read pairs by being hereby able to load the maximum number of samples onto a single flow cell without losing any sequencing information. This is especially beneficial when sequencing resources like flow cells are limited. The suppliers on the other hand can use our off-target analysis to further optimize their bait panels.

BTW, the published study should be forwarded to the companies developing the bait panels. We already did this (at least partially).

In the introduction, authors wrote "probes complementary to the target DNA". I suggest adding cDNA as well (or replacing DNA by nucleic acids).

We replaced "DNA" with "nucleic acids".

Also in the intro, the mention of the bait panel names is redundant.

We eliminated the listing of the bait panels in the introduction.

Please, write "bait panels" or "capture panels" instead of just "baits" (throughout the ms) and ORF1a instead of 1A.

Both suggestions were implemented throughout the manuscript.

In the discussion, please add a dot in naming the SARS-Cov-2 lineages (e.g., B.1.1.7), and correct B.1.315 to B.1.351. What does "and Co." mean there?

We corrected the notation of the SARS-CoV-2 lineages. "And Co." was NOT mentioned in context with the lineages but in the context of existing amplicon-based protocols. Nevertheless, we changed "ARTIC and Co." to "ARTIC and follow-up designs by various suppliers".

Please arrange the table columns in Table 2 according to Table 1 (as results shown in the Figures).

We changed the columns of Table 2 according to Table 1.

The manuscript should be reworked regarding comma/blank placement (e.g., first sentence of the second introductions' paragraph), especially Table 1 (i.e., use commas as separators for the read numbers).

We used commas as separators in Table 1 and fixed comma and blank placements.

In the methods, please add cities and countries for VWR, Roth, Thermo Fisher etc. as done for the other suppliers. In the "Sequence Capture by Hybridization" paragraph, please add the correct names of all tested bait panels (it makes more sense to mention them here than in the intro).

We edited the manuscript according to the suggestion of the reviewer.

Regarding supplementary Figures, the legends are very sparse and contain typos, legend for Fig. S4 is missing.

We complemented the missing figure legends.

Reviewer #2 (Comments for the Author):

This technical paper compares the performances of five different capture baits for SARS-CoV-2 sequencing. Such comparison has not been done yet and is important for laboratories seeking the best method to sequence SARS-CoV-2.

However, the main flaw of the study is the use of RNA pools with only 1 isolate of SARS-CoV-2 instead of patient's samples with large diversity of SARS-CoV-2. The analysis of non-

targets reads show that these pools do not necessarily reflect real patient samples, with non-targets reads being mostly rRNA in pools but not in patients samples.

Our study compares the general performance of the different capture kits. To get comparable results we had to focus on standard RNA background and a single viral strain. We did not investigate the general mechanism behind the capture technique and its sensitivity to mutations. Nevertheless, the capture kits worked very well for the mentioned 240 patient samples collected from May 2020 till March 2021, which were additionally derived from different countries of at least two continents. These patient samples reflect a broad diversity of lineages and lots of accumulated sample-specific mutations. Moreover, the catch and pooling of all these sequencing runs were based on our estimation of required reads published in this study. We did not observe any decrease in capture efficiency except for those samples with an unusual high amount of human background RNA. In summary, we can assure the reviewer that our observed data is applicable for patient samples of various origins and SARS-CoV-2 lineages.

RNA extracted from patient's samples usually have low RIN (~2). What was the RNA quality of the input pool? Does RNA quality impact the results?

We analyzed the RNA input pools and some patient samples with the Fragment Analyzer. As expected by the reviewer, the patient samples showed an RNA quality number (RQN) of 2.0–3.4, while the input pools showed an RQN of ~10, which is due to the high concentration of industrially produced human reference RNA. We didn't do a study with various input pools of different RQN qualities, but we can assure you from our sequencing experience of SARS-CoV-2 patient samples (not only the mentioned 240 patient samples) that library preparation as well the enrichment strategy work even with the low quality input.

More importantly, there are no details about the SARS-CoV-2 isolate used for the pools. Where does it come from? Which lineage does it belong to?

The SARS-CoV-2 isolate was derived from a patient sample and belongs to lineage B1. We added this information in the Methods part.

In the introduction and the discussion, the authors argue that capture-based approaches perform better than amplicon-based approaches for divergent viruses, however the paper does not evaluate whether this is similarly true for all tested baits. i.e do all baits capture equally well divergent SARS-CoV-2?

Unfortunately, the reviewer misinterpreted us. We never said that the capture-based approaches perform better than the amplicon-based approaches. We merely compare the disadvantage of the amplicon-based method, which is the risk of sequencing gaps due to mismatches between the amplicon primers and the continuously mutating viral target RNA with the advantage of the capture-based approach to overcome this issue. This is a general feature of all capture baits and is based on the design of all bait panels, as they are composed of 80-120 nt long nucleic acid fragments, which will still bind to the viral RNA, even in the case of several mutations. We added table 4 summarizing the characteristics of the various bait panels in the manuscript in the method section. Nevertheless, the reviewer has a valid point, which we addressed in the discussion as proposed by editor Rachel Mackelprang.

Other comments:

Fig 3 and 4: Please add error bars and statistical analysis to show whether capture efficiency is reproducible within the triplicates?

The request of the reviewer is unfortunately not possible to implement for Figure 3, because the triplicates were combined into one pool before the catch and, hence exited the capturing procedure as one pool. Therefore, no discrimination within the triplicates is possible in that state by ddPCR and hence no add error bars can be added. Concerning Figure 4, error bars would be barely visible due to the logarithmic scale of the y-axis. Instead, we decided to add a Suppl. Table S5 with the mean and standard deviation values used for plotting the figures. The average standard deviations are between 0.003 and 0.009, and hence far too small to plot distinguishable error bars.

Fig 4 and page 5: The analyses are made on reads "corrected for PCR duplicates". What is the impact of removing PCR duplicates? This is not done systematically in targeted sequencing.

Since we filtered out duplicated reads before we mapped the reads to the reference genome, we had to correct our estimated number of required reads by the duplication ratio, afterwards. Otherwise, we would get underestimated numbers. Nevertheless, we added statistics about PCR duplication ratio in Suppl. Table S6 and number of PCR cycles of each target enrichment kit in Table 4. In addition, we added a short paragraph to the discussion explaining that there is no direct correlation between the number of PCR pre- and post-enrichment cycles and duplication ratio.

Table 1: Number of reads needed for covering full-length SARS-CoV-2 at 20X should be calculated using minimal coverage depth, rather than median coverage depth.

Using the minimal coverage depth is mostly not applicable because it requires that all bases are covered, which was mostly not the case for pool 1-3. Hence, we have changed the formula to estimate the number of reads by using the overall mapping ratio rather than the coverage depth. The result is comparable to the previously provided numbers and we were able to estimate the numbers for the remaining pools, which have a median coverage of zero.

Table 1: it seems than 1 asterisk is missing for Pool 2 result using Twist Respiratory Panel

Since we have changed the formula to estimate the number of reads, we provide a new definition of the asterix. Now, it means that the number of reads required to cover the whole SARS-CoV-2 genome exceeds the number of sequencing reads obtainable from a single Illumina MiSeq v2 flow cell.

July 12, 2021

Dr. Mathias C. Walter
Bundeswehr Institute of Microbiology
Munich
Germany

Re: mSystems00392-21R1 (Catching SARS-CoV-2 by sequence hybridization: a comparative analysis)

Dear Dr. Mathias C. Walter:

Your manuscript has been accepted, and I am forwarding it to the ASM Journals Department for publication. For your reference, ASM Journals' address is given below. Before it can be scheduled for publication, your manuscript will be checked by the mSystems senior production editor, Ellie Ghatineh, to make sure that all elements meet the technical requirements for publication. She will contact you if anything needs to be revised before copyediting and production can begin. Otherwise, you will be notified when your proofs are ready to be viewed.

As an open-access publication, mSystems receives no financial support from paid subscriptions and depends on authors' prompt payment of publication fees as soon as their articles are accepted. =

Publication Fees:

We recognize that the video files can become quite large, and so to avoid quality loss ASM suggests sending the video file via <https://www.wetransfer.com/>. When you have a final version of the video and the still ready to share, please send it to Ellie Ghatineh at eghatineh@asmusa.org.

Sincerely,

Rachel Mackelprang
Editor, mSystems

Journals Department
Figure S4: Accept
Figure S1: Accept
Table S6: Accept
Table S2: Accept
Figure S3: Accept
Table S4: Accept
Table S1: Accept
Table S3: Accept
Table S5: Accept
Figure S2: Accept